# The Wellbeing Thermometer in Patients with Colorectal Cancer: A Validation Study

**DOI:** 10.3390/diseases12110280

**Published:** 2024-11-05

**Authors:** Marios Adamou, Okezie Uche-Ikonne, Konstantinos Kamposioras

**Affiliations:** 1School of Human and Health Sciences, University of Huddersfield, Queensgate, Huddersfield HD1 3DH, UK; 2The Christie NHS Foundation Trust, Manchester M20 4BX, UK

**Keywords:** GAD-7, PHQ-9, wellbeing thermometer, WHO-5

## Abstract

Background: Wellbeing is a valuable outcome with benefits for patients and the wider healthcare system. Different instruments are available to measure wellbeing; however, each has its own limitations. Existing wellbeing models focus mostly on a single aspect (e.g., social). The Wellbeing Thermometer (WbT) was developed based on a more holistic framework for wellbeing. Objective: The aim of this research was to validate the WbT on patients receiving treatments using a sample of patients with colorectal cancer in England. Methods: A survey, including GAD-7, PHQ-9, WHO-5, and WbT, was administered to two independent cohorts of adult patients diagnosed with colorectal cancer. The WbT consists of 25 questions/items: 5 for each domain of health, thoughts, emotions, spiritual, and social. We investigated the psychometric properties of the WbT to characterize item difficulty, discrimination, and reliability. Pearson’s correlation coefficient was used to compare WbT scores to those from other validated tools. A multivariable logistic model explored associations between WbT domains and other validated tools. Results: Cronbach’s alpha for WbT was 0.872 (95% confidence interval: 0.829–0.902), indicating good internal consistency. The item difficulty for WbT showed low scores for questions 6 (0.39) and 9 (0.49) and high scores for the other items. Item 3 in the health subgroup indicated weak discrimination towards the health item score (r = 0.35) and no discrimination towards the total score (r = 0.03). Item 1 in the spiritual subgroup showed weak discrimination towards the spiritual item score (r = 0.37). The WbT showed moderate to strong correlation with all other validated tools (r range: GAD-7, −0.49 to −0.77; PHQ-9, −0.69 to −0.83; WHO-5, 0.66 to 0.85). For Cohort 1, the WbT thought domain was associated with GAD-7 (*p* = 0.004) and WHO-5 (*p* = 0.002), and the health domain was associated with PHQ-9 (*p* = 0.014). For Cohort 2, the WbT thought domain was associated with GAD-7 (*p* = 0.02), the health domain was associated with WHO-5 (*p* = 0.02), and the emotion domain was associated with WHO-5 (*p* = 0.02). Conclusions: The WbT is a valid tool for assessing wellbeing in patients with colorectal cancer. The WbT may be a useful addition to both clinical practice and future research and may help shed light on a new area with regards to patients with cancer, specifically how they feel and function. This will ultimately increase wellbeing and reduce suffering.

## 1. Introduction

Wellbeing is a complex, multidimensional concept that lacks a universally accepted definition. However, it is generally understood to encompass “how people feel and how they function on a number of levels and how they evaluate their lives as a whole” [1]. The absence of a consensus definition, coupled with varying theoretical foundations, presents challenges in measuring wellbeing effectively [2]. This complexity is further compounded by disciplinary differences in conceptualizing wellbeing and its frequent conflation with related concepts such as happiness and wellness [2,3].

Despite these conceptual challenges, wellbeing remains a highly prioritized area of interest due to its significant implications for individual health outcomes and broader healthcare systems. The measurement and enhancement in subjective wellbeing have become key objectives in policy and governance, as traditional objective indicators (e.g., income) provide only a partial understanding of what constitutes a good life [2]. Comprehensive wellbeing assessment can inform the development of targeted health programs, workplace improvements, effective resource allocation, and the evaluation of public initiatives, ultimately contributing to the creation of a thriving society.

While numerous wellbeing assessment tools exist, there is no single, universally accepted measure that comprehensively captures all aspects of wellbeing, particularly in the context of cancer care. The Generalized Anxiety Disorder-7 (GAD-7), Patient Health Questionnaire-9 (PHQ-9), and World Health Organization-5 Well-Being Index (WHO-5) are widely used and validated instruments, each focusing on specific aspects of mental health and wellbeing. However, these tools, while valuable, do not provide a holistic view of wellbeing that encompasses physical, emotional, social, and spiritual domains.

The literature describes various wellbeing assessment tools, each with its own focus and limitations. The Warwick–Edinburgh Mental Well-being Scale [4] and the Personal Wellbeing Index [5] offer valuable insights but lack comprehensiveness in addressing all wellbeing domains. The WHO-5 [6] and the Physician Well-Being Index [7] focus on specific aspects of mental health or professional wellbeing but do not capture the full spectrum of wellbeing. Other notable tools measure various components such as mental health, affect, and psychological needs, yet often omit crucial elements like spiritual wellbeing [8,9].

Existing models of wellbeing typically concentrate on single aspects (e.g., psychological or social) and are often grounded in concepts of mental illness or life satisfaction rather than offering a unified wellbeing concept [1]. To address these limitations, a novel model of wellbeing was proposed, conceptualizing wellbeing domains as components of a complex adaptive system [10]. Based on this framework, the Wellbeing Thermometer (WbT) was developed to provide a more holistic approach, encompassing five key areas: health, thought, emotion, spiritual, and social domains [1].

The WbT presents a more comprehensive, theoretically grounded, and empowering approach to conceptualizing and measuring wellbeing compared to existing tools. It was designed to enable individuals to better understand and take control of their own wellbeing state, providing a self-directed approach that differs from the focus of some other instruments. A key distinction of the WbT is that it views wellbeing as an emergent state constituted by interrelated parts, while other models focus more on measuring individual components in isolation.

The WbT was developed by the authors to address the need for a more comprehensive, theoretically grounded approach to measuring wellbeing, particularly in clinical populations such as patients with cancer. It was created through a multidisciplinary effort involving a literature review, critical analysis of existing wellbeing tools, construction of a novel wellbeing framework, and development of relevant questions to enable measurement of wellbeing [1].

This study aims to validate the WbT in independent cohorts of adult patients with colorectal cancer at a tertiary care center in England. As this represents a novel framework, the questions have not been previously tested against established tools. The validation of this framework could significantly contribute to the exploration and understanding of wellbeing in clinical settings, particularly among patients with cancer.

## 2. Methods

### 2.1. Study Design, Setting, and Participants

In this study, patients were asked to complete the new tool (WbT) alongside other known validated psychometric tools described in the manuscript to correlate and validate the new tool according to the methodology outlined below. Data for these analyses were obtained from two independent cohorts of patients with cancer attending a large comprehensive cancer center in Northwest England (The Christie National Health Service [NHS] Foundation Trust). This was a convenience sample recruited as part of ‘Psychological Impact of COVID-19 on Patients with Solid Malignancies: A Single Institution Study (PICO-SM)’, a prospective longitudinal study designed to investigate the mental health burden of patients with cancer [11].

Details of this study were published previously [11]. In brief, consecutive patients were identified through clinic list review and recruited either between 7 and 28 April 2021 (Cohort 1, Timepoint 1) or between 6 December 2021 and 16 February 2022 (Cohort 2, Timepoint 1). Eligible patients were (a) aged 18 years or older; (b) diagnosed with colorectal cancer; and (c) able to fully comprehend the patient information sheet. Participants who lacked capacity were excluded. Participants completed the survey either onsite or via post. Participants who consented to be contacted again on a future date were approached six months after the day of completion of the initial survey for the completion of the survey again (i.e., 7 October 2021 to 21 January 2022 for Cohort 1, Timepoint 2). The follow-up survey for patients in Cohort 2 was not included due to the sample size being too small for meaningful analyses.

### 2.2. Ethical Considerations

This study was performed in line with the principles of the Declaration of Helsinki. Approval was granted by the Ethics Committee of the NHS Health Research Authority (protocol name: 21/WA/0021, IRAS ID: 292413), approval date: 26 February 2021. All patients provided informed consent to participate. All data were anonymized for the purpose of these analyses.

### 2.3. Survey Design

The administered survey was designed by a multidisciplinary team, which included oncologists, psychologists, psychiatrists, nurse specialists, and patient representatives. The survey included questions on patient demographics (i.e., gender, age, ethnicity, marital status) and clinical characteristics (i.e., previous diagnosis of mental health condition, perception of current status of cancer) and also comprised questions from tools including the Generalized Anxiety Disorder scale (GAD-7), Patient Health Questionnaire-9 (PHQ-9), WHO-5, and WbT [1,6,12,13,14].

### 2.4. Study Measures

#### 2.4.1. Wellbeing Thermometer

This tool consists of 25 questions; 5 for each domain of health, thoughts, emotions, spiritual, and social [1]. One point is given for each question answered positively in relation to the wellbeing state and zero points are given for negative answers relating to the wellbeing state. Together, the answers give an indication of a person’s wellbeing status; a total score of 25 indicates the best imaginable wellbeing. The WbT is currently under copyright; contact the authors for an approval of use.

#### 2.4.2. Validated Measures

Anxiety

The GAD-7 scale to measure the severity of self-reported anxiety within the past two weeks consists of seven questions, each scored according to response categories of ‘not at all’ (0 points), ‘several days’ (1 point), ‘more than half the days’ (2 points), and ‘nearly every day’ (3 points). A total score of 0–4, 5–9, 10–14, or >15 indicates minimal, mild, moderate, or severe anxiety, respectively [12]. This tool has been validated in various populations, including patients with cancer. Wong et al. (2019) and Esser et al. (2018) confirmed its utility as an effective screening tool for anxiety in oncology settings [15,16,17,18].

Depression

The PHQ-9 is a validated instrument for screening, diagnosing, monitoring, and measuring the severity of self-reported depression within the past two weeks [13,19,20]. It consists of nine questions, each scored according to response categories of ‘not at all’ (0 points), ‘several days’ (1 point), ‘more than half the days’ (2 points), and ‘nearly every day’ (3 points). A total score ranging from 0 to 5 represents mild depression, 6 to 10 represents moderate depression, 11 to 15 represents moderately severe depression, and 16 to 20 represents severe depression. The PHQ-9 has demonstrated reliability and validity in cancer populations. Studies by Ganz et al. (2021) and Park (2024) showed its effectiveness in identifying depressive symptoms among patients with cancer, particularly in breast cancer survivors [19,20,21,22].

Mental wellbeing

The WHO-5 for mental wellbeing is a short self-reported measure of current mental wellbeing, which consists of five statements, which respondents rate according to the following scale: ‘all of the time’ (5 points); ‘most of the time’ (4 points); ‘more than half of the time’ (3 points); ‘less than half of the time’ (2 points); ‘some of the time’ (1 point); and ‘at no time’ (0 points). The total raw score is multiplied by four to give the final score ranging from 0 (worst imaginable wellbeing) to 100 (best imaginable wellbeing) [14]. Reliability and validity have been confirmed in different patient populations [23,24,25,26,27,28,29]. While the WHO-5 has been validated in various populations, its specific use in patients with cancer has also been documented. Hoffman et al. (2012) used the WHO-5 in a study of patients with breast cancer, where baseline scores around 50 aligned with the established cut-off for screening depression, indicating its utility in identifying patients at risk for depressive symptoms in the context of cancer treatment [23,24,25,26,27,28,29,30].

### 2.5. Statistical Analyses

Data analyses were performed using R version 4.1.3 (R Core Team, Vienna, Austria, 2022). Only questionnaires with complete responses (i.e., where patients had completed all questions for all tools) were utilized for further analyses. Descriptive statistics were used to summarize patient demographic and clinical characteristics and scores for the different tools by cohort and timepoint. Continuous variables were expressed as means and standard deviations (SDs), and categorical variables were expressed as frequencies and percentages.

For the analysis of psychometric properties, item difficulty was assessed by calculating the proportion of responses indicative of positive wellbeing for each item. For example, in a question like “I feel optimistic about the future”, a response of “Yes” would be considered a “correct” answer in terms of positive wellbeing. Item discrimination was evaluated to determine how well each question distinguished between different levels of overall wellbeing.

Cohort 1 Timepoint 1 was used for the analysis of psychometric properties of the wellbeing tool as it had the largest sample size. The properties included item difficulty (i.e., how challenging each question is for the respondent), item discrimination (i.e., assesses how well individual questions distinguish the varying levels of wellbeing), and item reliability (i.e., consistency of WbT in measuring wellbeing). Item difficulty was measured by calculating the proportion of correct answers and item discrimination was estimated using the point biserial correlation for the item response and total score. We examined the discrimination between each item score within the health, social, mental, emotional, and spiritual subgroups, as well as the overall score. Discrimination was reported as follows: <0—Negative discrimination; 0–0.2—No discrimination; 0.2–0.4—Weak discrimination; and >0.4—Strong discrimination. Cronbach’s alpha was used to measure item reliability and consistency was reported as follows: 0.9 ≤ α—Excellent; 0.8 ≤ α < 0.9—Good; 0.7 ≤ α < 0.8—Acceptable; 0.6 ≤ α < 0.7—Questionable; 0.5 ≤ α < 0.6—Poor; and α < 0.5—Unacceptable.

Pearson’s correlation coefficient was used to compare the relationship between the WbT scores and scores from the GAD-7, PHQ-9, and WHO-5 tools for each cohort. A low correlation was defined as r < 0.45, a moderate correlation as 0.45 ≤ r < 0.7, and a strong correlation as r ≥ 0.70. To examine the consistency of responses, Pearson’s r coefficient was calculated for matched patients in Cohort 1 who completed the questionnaire at both timepoints.

To explore associations between the domains of the WbT (i.e., health, social, thoughts, emotions, spiritual) and other validated tools (i.e., GAD-7, PHQ-9, WHO-5), the results of the other tools’ scores were first divided into two groups using the following cut-off values:GAD-7 ≤ 5, cut-off for mild to severe anxiety [31];PHQ-9 ≤ 10, cut-off for moderate to severe depression [32];WHO-5 ≥ 50, cut-off for good wellbeing [6].

A multivariable logistics model was analyzed to generate the odds ratio coefficient. A *p*-value of <0.05 was considered statistically significant.

## 3. Results

### 3.1. Patients

Of 216 patients with colorectal cancer who participated in the PICO-SM study between 7 and 28 April 2021, 127 (59%) had complete responses and were included in this study (Cohort 1 Timepoint 1; Figure 1). Of 96 patients with colorectal cancer who completed the survey between 6 December 2021 and 16 February 2022, 57 (59%) had complete responses and were included in this study. Baseline demographics and clinical characteristics of the cohorts are presented in Table 1. The mean age of patients in Cohort 1 was 64 years, and 58% were male (n = 74/127). The most common underlying mental health conditions reported included anxiety (13%), psychosis (8%), and depression (5%). Fifty percent of respondents reported stable disease (n = 63/126). For Cohort 2, the mean age was 63 years and 68% were male. Again, anxiety (21%), psychosis (16%), and depression (5%) were the most common underlying mental health conditions. Stable disease was self-reported in 48% (n = 27/56).

### 3.2. Scores

Tool scores for Cohort 1 Timepoint 1 were as follows: GAD-7 mean, 3.72 (SD: 5.34); PHQ-9, 4.95 (5.33); WHO-5, 60.51 (23.30); and WbT, 18.8 (5.09). The scores for Cohort 2 Timepoint 1 were as follows: GAD-7 mean, 3.63 (SD: 4.35); PHQ-9, 4.81 (4.66); WHO-5, 59.93 (21.16); and WbT, 19.02 (5.07).

#### Comparison Between Cohorts

We use Welch’s T-test to compare the means of the total scores for the two cohorts; there was no statistical difference between the cohorts for any wellbeing measure (GAD-7: t = 0.208, *p* = 0.835; PHQ-9: t = 0.147, *p* = 0.884; WHO-5: t = 0.374, *p* = 0.709; WbT: t = −0.041, *p* = 0.967).

### 3.3. Psychometric Analysis

Questions 3, 7, 10, and 21 were the easiest to answer with >90% of respondents answering these items correctly (Table 2). Less than 50% of respondents answered questions 6 and 9 correctly, indicating that these questions were harder. The histogram of item difficulty scores displayed in Figure 2 is left-skewed, indicating that most items were easy for the respondents to answer.

Item discrimination against each subgroup is displayed in Table 3. Every item significantly discriminated the respondents based on each subgroup except for item 3 in the health subgroup and item 1 in the spiritual subgroup. The discrimination measure against the total score is displayed in Table 4. Seven items have weak discrimination, and seventeen items have a strong discrimination towards the total score. Item 3 has no discrimination of respondents towards their overall wellbeing score.

Cronbach’s alpha is 0.872 (0.829–0.902), indicating good internal consistency.

### 3.4. Correlation Analyses

For Cohort 1 Timepoint 1, the WbT was moderately negatively correlated with GAD-7 (r = −0.49) and PHQ-9 (r = −0.69) and moderately positively correlated with WHO-5 (r = 0.66) (Figure 3A). Results were similar when analyses were restricted to include only scores from individuals in Cohort 1 with data available for both timepoints (Online Resource 1). For Cohort 2 Timepoint 1, the WbT was strongly negatively correlated with GAD-7 (r = −0.7) and PHQ-9 (r = −0.75) and strongly positively correlated with WHO-5 (r = 0.74).

### 3.5. Multivariable Analyses

The thought domain in the WbT was significantly associated with GAD-7 cut-off 5 (OR = 0.49; *p* = 0.002) and WHO-5 (OR = 0.49; *p* = 0.002) for Cohort 1 Timepoint 1, and the health domain was significantly associated with PHQ-9 (OR = 0.48; *p* = 0.014; Table 5). For Cohort 2 Timepoint 1, the thought domain of the WbT was significantly associated with GAD-7 cut-off 5 (OR = 0.37; *p* = 0.02), the health domain was significantly associated with WHO-5 (OR = 0.33; *p* = 0.02), and the emotion domain was significantly associated with WHO-5 (OR = 0.22; *p* = 0.02; Table 5).

## 4. Discussion

The WbT is a new tool devised to address the limitations of existing instruments available to measure wellbeing [1]. To our knowledge, this is the first study to compare the WbT with other validated tools in any patient cohort, specifically focusing on patients with colorectal cancer.

The use of validated tools such as the GAD-7, PHQ-9, and WHO-5 alongside the WbT provides a robust framework for assessing the validity of the new instrument. These tools have been extensively validated in cancer populations, with studies demonstrating their effectiveness in screening for anxiety, depression, and overall wellbeing in oncology settings [18,21,22,30]. The strong correlations observed between the WbT and these established measures suggest that the WbT captures important aspects of mental health and wellbeing relevant to patients with cancer.

Psychometric analyses showed that most questions were easy for respondents to answer. Two questions (questions 6 and 9) appeared to be somewhat more challenging and should be reviewed, as should question 3, as indicated by discrimination analyses. The aim would be to ensure that these questions are appropriately challenging (neither too easy nor too difficult) and capable of distinguishing between different levels of wellbeing among respondents. Adjustments might include rephrasing questions, changing their format, or even replacing them with more effective items.

The WbT showed consistent moderate to strong correlation with the other tools in patients with colorectal cancer. Additionally, several domains of the WbT were significantly associated with the other validated measures. We found that the WbT negatively correlated with GAD-7 and PHQ-9 for all cohorts and timepoints; i.e., a higher wellbeing correlated with a lower score for anxiety and depression, respectively. Similarly, the WbT positively correlated with WHO-5 for all cohorts and timepoints. Thus, the WbT had the same direction of correlation in all three validation cohorts for the different tools. There was some variation in the association of different domains of the WbT with other validated tools. For example, the thought domain of the WbT was associated with GAD-7 in both Cohort 1 Timepoint 1 and Cohort 2 Timepoint 1; however, the health domain was associated with PHQ-9 in the former and with WHO-5 in the latter. These differences between cohorts are likely to reflect the fact that patients in different cohorts are at different points in their cancer journey, which is reflected in differences in their wellbeing. 

The WHO-5 tool is commonly used in oncology to assess patients’ wellbeing [30,33]. It is a short self-reported measure derived from the WHO-10, but its layout follows that of the Major Depression Inventory, which measures the WHO/ICD-10 symptoms of major depressive disorder and not wellbeing [6,18]. Thus, the tool is limited by the lack of certain domains. The inclusion of five domains in the WbT can help to identify areas where individuals might need more support and tailor interventions accordingly. Further strengths of this study include the validation of the WbT through three different tools (i.e., GAD-7, PHQ-9, WHO-5) and at three different timepoints. The WbT is easy and quick to complete, and hence could be used in clinical settings and incorporated in electronic patient records. Although the WbT is not expected to detect disease status, it could highlight patients who may require further support.

The WbT’s potential extends beyond its use as a clinical tool. By providing a more comprehensive assessment of wellbeing, it could inform the development of more holistic care plans, guide resource allocation in healthcare systems, and contribute to public health initiatives aimed at improving overall quality of life for patients with cancer. Furthermore, its applicability to other patient groups and even non-patient populations opens avenues for broader research into wellbeing across various contexts.

A limitation of the study is the small sample size and number of missing responses. However, this is expected in patients with colorectal cancer where approximately one third report progressive disease. Despite being conducted in a convenience sample of patients with colorectal cancer, the WbT should not be regarded as disease-specific and is applicable to other cancer groups as oncology patients share similar concerns.

This tool is designed to bring to light new understanding regarding the wellbeing of patients, which is grounded on a holistic framework and not disease-specific. Both mental and physical conditions should be considered in the overall management of patients; otherwise, patients are being deprived of support opportunities, which could in turn increase wellbeing and reduce their suffering [21]. The WbT may be a useful addition to both clinical practice and future research. It can be used specifically in oncology as an additional outcome measure to assess how patients are coping with treatment, for example. Measuring wellbeing not only helps to evaluate the impact of services on people’s lives, but also aspects of their lives people feel most dissatisfied with. The latter can be used to help tailor services to meet their needs. The WbT can also be used beyond the oncology setting in other patient and non-patient groups.

## 5. Conclusions

The WbT appears to be a valid instrument for assessing wellbeing in patients with cancer. We believe that the WbT could be used as part of a holistic assessment to support patients under oncological management. However, further testing and replication using a larger sample size are recommended to increase measurement accuracy and reliability. It will be useful to investigate various thresholds or meaningful ranges for the WbT to measure variations in wellbeing. Future research should also explore the applicability of the WbT in different cancer types and stages, as well as its potential use in longitudinal studies to track changes in wellbeing throughout the cancer journey.

## Figures and Tables

**Figure 1 diseases-12-00280-f001:**
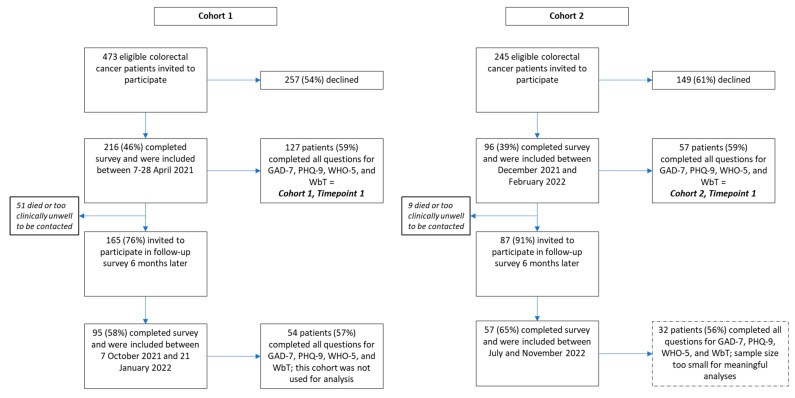
Patient flow diagram.

**Figure 2 diseases-12-00280-f002:**
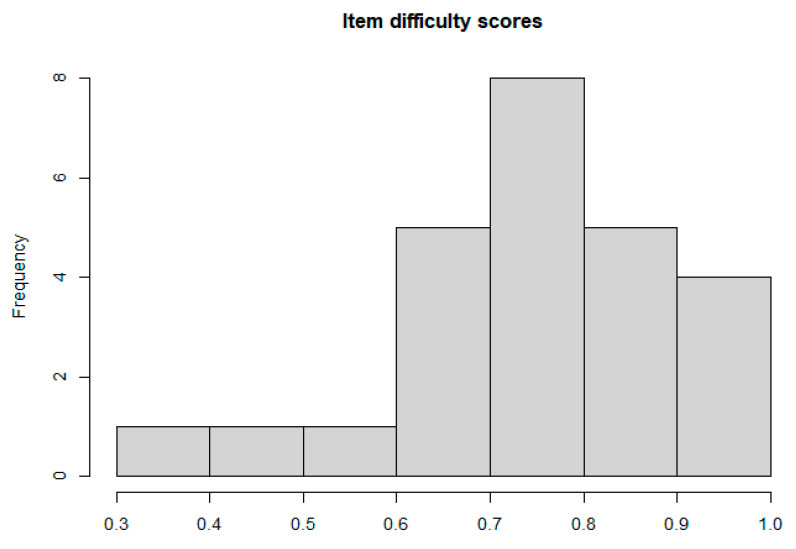
Histogram of item difficulty scores.

**Figure 3 diseases-12-00280-f003:**
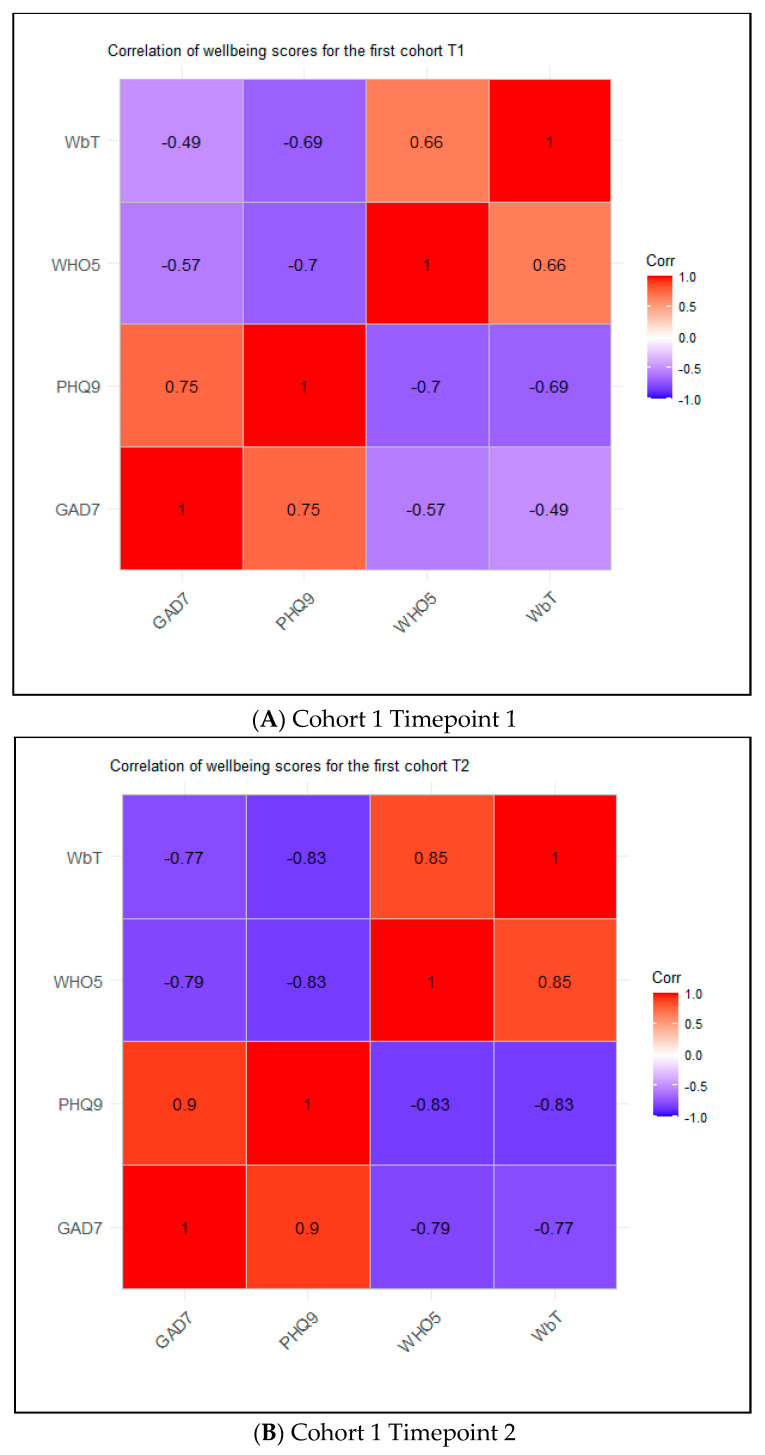
Correlogram for cohorts.

**Table 1 diseases-12-00280-t001:** Baseline demographics and clinical characteristics of patient cohorts.

	Cohort 1 (n = 127)	Cohort 2 (n = 57)
Gender, n (%)		N = 56
Male	74 (58.3)	38 (67.9)
Age, mean ± SD (years)	64 ± 9.59	63 ± 11.36
Ethnicity, n (%)		N = 56
White/White, British	117 (92.1)	53 (94.6)
Marital status, n (%)		N = 56
Single/Divorced/Separated/Widowed	41 (32.3)	10 (17.9)
In a relationship/ Married/In civil partnership	86 (67.7)	46 (82.1)
Children, n (%)	N = 126	N = 56
Yes	100 (79.4)	45 (80.4)
No	26 (20.6)	11 (19.6)
Lives alone, n (%)	N = 126	N = 56
Yes	31 (24.6)	6 (10.7)
No	95 (75.4)	50 (89.3)
Previous/underlying diagnosis of mental health condition, n (%)		
Anxiety	17 (13.4)	12 (21)
Depression	6 (4.7)	3 (5.3)
Panic attacks	3 (2.4)	0
Anorexia	3 (2.4)	0
Psychosis	10 (7.9)	9 (15.8)
Bulimia	3 (2.4)	0
Social phobia	3 (2.4)	0
Attention deficit disorder	3 (2.4)	0
Obsessive compulsive disorder	3 (2.4)	1 (1.7)
Autism	3 (2.4)	0
Post-traumatic stress disorder	1 (0.8)	2 (3.5)
Alcohol/drugs	3 (2.4)	0
Bipolar disorder	1 (0.8)	0
Personality disorder	1 (0.8)	0
Other	1 (0.8)	0
None of the above	97 (76.4)	36 (63.2)
Self-reported perception of current status of cancer, n (%)	N = 126	N = 56
Stable disease	63 (50)	27 (48.2)
Progressive disease	43 (34.1)	19 (33.9)
Unknown	20 (15.9)	10 (17.9)

**Table 2 diseases-12-00280-t002:** Item difficulty score for WbT.

Question	Difficulty
Q1	0.71
Q2	0.64
Q3	0.92
Q4	0.74
Q5	0.64
Q6	0.39
Q7	0.96
Q8	0.86
Q9	0.49
Q10	0.92
Q11	0.82
Q12	0.8
Q13	0.79
Q14	0.56
Q15	0.64
Q16	0.77
Q17	0.69
Q18	0.69
Q19	0.88
Q20	0.74
Q21	0.95
Q22	0.8
Q23	0.81
Q24	0.78
Q25	0.82

**Table 3 diseases-12-00280-t003:** Item discrimination for each subgroup of WbT.

	Question 1	Question 2	Question 3	Question 4	Question 5
Health	0.64	0.57	0.35	0.6	0.71
Social	0.78	0.4	0.54	0.8	0.56
Thoughts	0.72	0.81	0.73	0.53	0.62
Emotions	0.72	0.73	0.69	0.54	0.65
Spiritual	0.37	0.8	0.79	0.86	0.8

**Table 4 diseases-12-00280-t004:** Item discrimination score for WbT.

Item	Discrimination
Q1	0.38
Q2	0.41
Q3	0.08
Q4	0.58
Q5	0.53
Q6	0.39
Q7	0.22
Q8	0.38
Q9	0.46
Q10	0.37
Q11	0.49
Q12	0.74
Q13	0.68
Q14	0.38
Q15	0.53
Q16	0.67
Q17	0.54
Q18	0.43
Q19	0.59
Q20	0.63
Q21	0.25
Q22	0.65
Q23	0.64
Q24	0.7
Q25	0.56

**Table 5 diseases-12-00280-t005:** OR coefficients and *p*-values for the multivariable models by cohort.

		Cohort 1 Timepoint 1		Cohort 2 Timepoint 1	
Wellbeing Tool	Descriptors	OR Coefficients	*p*-Value	OR	*p*-Value
GAD-7 (cut-off 5)	Health	0.798	0.304	0.682	0.282
	Social	0.93	0.732	0.505	0.134
	Thoughts	0.488	0.002	0.371	0.02
	Emotions	0.974	0.907	0.461	0.09
	Spiritual	0.959	0.841	2.221	0.05
PHQ-9 (cut-off 10)	Health	0.484	0.014	0.866	0.871
	Social	0.696	0.216	0.032	0.185
	Thoughts	0.55	0.063	0.061	0.321
	Emotions	0.578	0.053	0.015	0.165
	Spiritual	1.61	0.101	0.108	0.214
WHO-5 (cut-off 50)	Health	0.768	0.232	0.334	0.02
	Social	0.763	0.21	0.373	0.09
	Thoughts	0.493	0.002	1.176	0.71
	Emotions	0.921	0.725	0.224	0.02
	Spiritual	0.806	0.309	0.707	0.391

GAD-7: Generalized Anxiety Disorder scale; OR: odds ratio; PHQ-9: Patient Health Questionnaire-9; WHO-5: World Health Organization-Five Well-Being Index.

## Data Availability

The data sets generated during and/or analyzed during this study are available from the corresponding author on reasonable request.

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
