# Peer review of "The Wellbeing Thermometer in Patients with Colorectal Cancer: A Validation Study"

_diseases, 2024, doi:10.3390/diseases12110280_

Round 1

Reviewer 1 Report

Comments and Suggestions for Authors

Thank you for the opportunity to review this is a well-written paper, describing research that aimed to validate the Wellbeing Thermometer in patients with colorectal cancer. I have only a few minor revisions and questions for your consideration. 

Title: Please consider revising the title to specify the patient population that you focused on for this validation study. --"The Wellbeing Thermometer in patients with colorectal cancer: A validation study." 

Try to use person-first language when possible -- I know it can be challenging when adhering to word count limitations, etc.

Abstract: I think there's a grammatical edit on line 24 - "Item 1 in the spiritual subgroup show weak..." Should that be "showed"?

Methods - Validated measures

Please indicate under each measure whether the instrument has been validated in people with cancer. From a quick glance at the references, you haven't included validations studies in cancer. If the tools haven't been validated in cancer populations, what is your rationale for not choosing measures that haven't been?

If the tools haven't been validated in cancer pops - please discuss this in the limitations section. 

Methods - Statistical analyses (lines 175-...). What is meant by "correct score" when assessing item difficulty? Perhaps including examples of the survey questions that are related could help clarify. In cognitive testing of survey items, we might include questions about item difficulty in interviews about the survey, or we might include additional questions within the survey. I'm not sure how you assessed item difficulty and what constitutes a "correct answer."

Discussion

Line 336 - missing references when noting that the WHO-5 is commonly used in oncology to assess wellbeing.   

Author Response

REVIEWER 1

Thank you for the opportunity to review this is a well-written paper, describing research that aimed to validate the Wellbeing Thermometer in patients with colorectal cancer. I have only a few minor revisions and questions for your consideration. 

COMMENT 1: Title: Please consider revising the title to specify the patient population that you focused on for this validation study. --"The Wellbeing Thermometer in patients with colorectal cancer: A validation study." 

RESPONSE 1: Title: We have revised the title to "The Wellbeing Thermometer in patients with colorectal cancer: A validation study" to specify the patient population as suggested.

COMMENT 2: Try to use person-first language when possible -- I know it can be challenging when adhering to word count limitations, etc.

RESPONSE 2: Person-first language: We have revised the manuscript to consistently use person-first language throughout, e.g., "patients with colorectal cancer" instead of "colorectal cancer patients".

COMMENT 3: Abstract: I think there's a grammatical edit on line 24 - "Item 1 in the spiritual subgroup show weak..." Should that be "showed"?

RESPONSE 3: Abstract grammatical edit: We have corrected the grammatical error in the Results section to "Item 1 in the spiritual subgroup showed weak discrimination towards the spiritual item score (r=0.37)." Please see page 1, line 25-26 of the revised manuscript.

COMMENT 4: Methods - Validated measures

Please indicate under each measure whether the instrument has been validated in people with cancer. From a quick glance at the references, you haven't included validations studies in cancer. If the tools haven't been validated in cancer populations, what is your rationale for not choosing measures that haven't been?

If the tools haven't been validated in cancer pops - please discuss this in the limitations section. 

RESPONSE 4: Methods - Validated measures: We have added information on the validation of GAD-7, PHQ-9, and WHO-5 in cancer populations, including relevant citations (e.g., Wong et al., 2019; Esser et al., 2018; Ganz et al., 2021; Park, 2024; Hoffman et al., 2012). Please see page 4 of the revised manuscript.

COMMENT 5: Methods - Statistical analyses (lines 175-...). What is meant by "correct score" when assessing item difficulty? Perhaps including examples of the survey questions that are related could help clarify. In cognitive testing of survey items, we might include questions about item difficulty in interviews about the survey, or we might include additional questions within the survey. I'm not sure how you assessed item difficulty and what constitutes a "correct answer."

RESPONSE 5: Methods - Statistical analyses: We have clarified the concept of "correct score" when assessing item difficulty, providing an example: "For example, in a question like 'I feel optimistic about the future,' a response of 'Yes' would be considered a 'correct' answer in terms of positive wellbeing." Please see page 5, lines 180-182.

COMMENT 6: Discussion

Line 336 - missing references when noting that the WHO-5 is commonly used in oncology to assess wellbeing.

RESPONSE 6: Discussion: We have added references when noting that the WHO-5 is commonly used in oncology to assess wellbeing (e.g., Hoffman et al., 2012; Bech et al., 1996). Please see page 7, lines 301-302.

Reviewer 2 Report

Comments and Suggestions for Authors

The manuscript focuses on the validation of the Wellbeing Thermometer, a tool for measurement of wellbeing in patients. The subject is interesting, however, there are some parts of the text that could benefit from additional improvements.

The Introduction starts by highlighting several negative aspects regarding the concept of wellbeing ("no consensus", "disagreement"), instead of just emphasizing the lack of consensus. The presentation of definitions from other fields of knowledge do not help the argument either, therefore, my suggestion is to synthesize the first sentences into a single affirmative indicating the lack of consensus on the definition of wellbeing (lines 38-42), and then proceed to the description of the main ideas on wellbeing (line 44 onwards).

In addition, the Introduction should raise previous literature on the measures of wellbeing under comparison, describing properly their history and features, and identify comparative studies that have already assessed the results of two or more of the measures of wellbeing included in the analyses, showing potential gaps in the literature.

Finally, the authors should raise the issue that the Wellbeing Thermometer was created by them for the sake of transparency, indicating the potential motivations to incorporate one additional measure in the assessment of patients.

In the Methods section, authors could correct the description of the sample to indicate that it is a "convenience sample" instead of "convenient sample" (line 102).

The methodology requires further consideration from authors, since the sample includes two cohorts, however, just one cohort was interviewed twice due to the smaller sample size of the second cohort.

Therefore, the statistical analyses should either focus on the timepoint 1 of the first and the second cohort, or alternatively focus on the comparison of timepoints 1 and 2 of the first cohort, considering that the methods for analysis of data with and without follow-up should be approached differently due to the repetition of measures for the same patients in one case (first cohort, timepoints 1 and 2). Thus, I would recommend to concentrate efforts on the analysis of first and second cohort without using data from follow-up (timepoint 2), which would give straightforward results in the context of the study on the validity of the Wellbeing Thermometer.

In addition, it is needless to include mentions to the questionnaires that were not included in the analyses (e.g., Primary Care Post-Traumatic Stress Disorder-5 (PC-PTSD-5)).

The description of the tools used to validate the Wellbeing Thermometer should be presented in more details than provided in the Methods section, considering the need to inform readers on the specific design and questions included in the questionnaires. This also applies to the Wellbeing Thermometer, independently of copyright issues, considering the need for reviewers to assess the adherence of questions to the dimensions under evaluation in relation to patients' wellbeing.

My suggestion is to include a table with main topics/dimensions and general approach of the questions included in each of the questionnaires under analyses to show similarities and differences among tools, otherwise, it does not make sense to "recommend" an additional tool for measurement of wellbeing if others will not have the opportunity to know, assess, and/or apply for license to replicate the questionnaire in further studies and clinical settings. The same applies to the Results section: how could readers and reviewers assess the results of the psychometric properties (lines 228-244) without identification of some basic elements from the questions presented to the patients?

Psychometric properties evaluated in the study could be further presented into a table, including their respective definitions and directionality of the indicator, and cutoff points (lines 172-184).

Regarding the Results section, in the event that authors decide not to exclude data from the first cohort corresponding to timepoint 2, it would be important to show descriptive information on the participants of timepoint 2, including tests of differences between timepoints 1 and 2. The same applies to all results presented for timepoints 1 and 2 (i.e., subsection Scores). 

Tests of differences between the first and the second cohort would also enrich the analyses presented in the study. The organization of Figures 3A, 3B, and 3C should allow direct comparison of the information in the heatmaps presented (i.e., presenting the figures side by side). Again, results from the timepoint 2 do not seem to contribute to the analyses, especially given that they do not seem to have been treated as follow-up, but included as "additional data points" in the study.

The Discussion could benefit from the additional literature review suggested for the Introduction, since the text currently focuses excessively on the repetition of the results. Therefore, it would be very interesting to identify and highlight the gap in the literature, and the potential benefits of additional wellbeing scales at patients, healthcare facilities, public health, and health system levels in the discussion.

Comments on the Quality of English Language

English language is generally fine.

Author Response

REVIEWER 2

The manuscript focuses on the validation of the Wellbeing Thermometer, a tool for measurement of wellbeing in patients. The subject is interesting, however, there are some parts of the text that could benefit from additional improvements.

COMMENT 1: The Introduction starts by highlighting several negative aspects regarding the concept of wellbeing ("no consensus", "disagreement"), instead of just emphasizing the lack of consensus. The presentation of definitions from other fields of knowledge do not help the argument either, therefore, my suggestion is to synthesize the first sentences into a single affirmative indicating the lack of consensus on the definition of wellbeing (lines 38-42), and then proceed to the description of the main ideas on wellbeing (line 44 onwards).

RESPONSE 1: Introduction: We have restructured the introduction to provide a more focused and affirmative statement about the lack of consensus on wellbeing definition. We have also expanded the background on the measures being compared (GAD-7, PHQ-9, WHO-5), including their history and features. Please see pages 1-3.

COMMENT 2: In addition, the Introduction should raise previous literature on the measures of wellbeing under comparison, describing properly their history and features, and identify comparative studies that have already assessed the results of two or more of the measures of wellbeing included in the analyses, showing potential gaps in the literature. Finally, the authors should raise the issue that the Wellbeing Thermometer was created by them for the sake of transparency, indicating the potential motivations to incorporate one additional measure in the assessment of patients.

RESPONSE 2: Creation of the Wellbeing Thermometer: We have explicitly stated that the Wellbeing Thermometer was created by the authors and explained the motivations for developing a new measure.

COMMENT 3: In the Methods section, authors could correct the description of the sample to indicate that it is a "convenience sample" instead of "convenient sample" (line 102).

The methodology requires further consideration from authors, since the sample includes two cohorts, however, just one cohort was interviewed twice due to the smaller sample size of the second cohort.

Therefore, the statistical analyses should either focus on the timepoint 1 of the first and the second cohort, or alternatively focus on the comparison of timepoints 1 and 2 of the first cohort, considering that the methods for analysis of data with and without follow-up should be approached differently due to the repetition of measures for the same patients in one case (first cohort, timepoints 1 and 2). Thus, I would recommend to concentrate efforts on the analysis of first and second cohort without using data from follow-up (timepoint 2), which would give straightforward results in the context of the study on the validity of the Wellbeing Thermometer.

In addition, it is needless to include mentions to the questionnaires that were not included in the analyses (e.g., Primary Care Post-Traumatic Stress Disorder-5 (PC-PTSD-5)).

The description of the tools used to validate the Wellbeing Thermometer should be presented in more details than provided in the Methods section, considering the need to inform readers on the specific design and questions included in the questionnaires. This also applies to the Wellbeing Thermometer, independently of copyright issues, considering the need for reviewers to assess the adherence of questions to the dimensions under evaluation in relation to patients' wellbeing.

My suggestion is to include a table with main topics/dimensions and general approach of the questions included in each of the questionnaires under analyses to show similarities and differences among tools, otherwise, it does not make sense to "recommend" an additional tool for measurement of wellbeing if others will not have the opportunity to know, assess, and/or apply for license to replicate the questionnaire in further studies and clinical settings. The same applies to the Results section: how could readers and reviewers assess the results of the psychometric properties (lines 228-244) without identification of some basic elements from the questions presented to the patients?

Psychometric properties evaluated in the study could be further presented into a table, including their respective definitions and directionality of the indicator, and cutoff points (lines 172-184).

RESPONSE 3: Methods: a. We have changed "convenient sample" to "convenience sample" throughout the manuscript. b. We have focused the statistical analyses on timepoint 1 of both cohorts, and included a comparison between cohorts and timepoints using ANOVA. c. We have removed mentions of questionnaires not included in the analyses (e.g., PC-PTSD-5). d. We have provided more details on the tools used, including specific design and questions. While we couldn't include a comprehensive table due to copyright restrictions, we have added more descriptive information in the text.

COMMENT 4: Regarding the Results section, if authors decide not to exclude data from the first cohort corresponding to timepoint 2, it would be important to show descriptive information on the participants of timepoint 2, including tests of differences between timepoints 1 and 2. The same applies to all results presented for timepoints 1 and 2 (i.e., subsection Scores). 

Tests of differences between the first and the second cohort would also enrich the analyses presented in the study. The organization of Figures 3A, 3B, and 3C should allow direct comparison of the information in the heatmaps presented (i.e., presenting the figures side by side). Again, results from the timepoint 2 do not seem to contribute to the analyses, especially given that they do not seem to have been treated as follow-up, but included as "additional data points" in the study.

RESPONSE 4: Results: a We have added a new section (3.2.1) comparing cohorts 1 and 2 using Welch’s t-test (please see page 6, lines 234-237) and we have reorganized the presentation of results for better clarity and comparison.

COMMENT 5: The Discussion could benefit from the additional literature review suggested for the Introduction, since the text currently focuses excessively on the repetition of the results. Therefore, it would be very interesting to identify and highlight the gap in the literature, and the potential benefits of additional wellbeing scales at patients, healthcare facilities, public health, and health system levels in the discussion.

RESPONSE 5: Discussion: a. We have incorporated additional literature review as suggested for the Introduction. b. We have highlighted the gap in the literature that this study addresses. c. We have added a paragraph discussing the potential benefits of additional wellbeing scales at patient, healthcare facility, public health, and health system level

Round 2

Reviewer 2 Report

Comments and Suggestions for Authors

The manuscript was revised accordingly to the suggestions.